# The Potential of Proteolytic Chimeras as Pharmacological Tools and Therapeutic Agents

**DOI:** 10.3390/molecules25245956

**Published:** 2020-12-16

**Authors:** Bernat Coll-Martínez, Antonio Delgado, Bernat Crosas

**Affiliations:** 1Molecular Biology Institute of Barcelona (IBMB), Spanish National Research Council (CSIC), Barcelona Science Park, 08028 Barcelona, Spain; bcobmc@ibmb.csic.es; 2IQS School of Engineering, Universidad Ramon Llull, Via Augusta 390, 08017 Barcelona, Spain; 3Research Unit on Bioactive Molecules (RUBAM), Department of Biological Chemistry, Institute for Advanced Chemistry of Catalonia (IQAC-CSIC), Jordi Girona 18-26, 08034 Barcelona, Spain; delgado.rubam@gmail.com; 4Unit of Pharmaceutical Chemistry (Associated Unit to CSIC), Department of Pharmacology, Toxicology and Medicinal Chemistry, Faculty of Pharmacy and Food Sciences, University of Barcelona (UB), Avda. Joan XXIII 27-31, 08028 Barcelona, Spain

**Keywords:** chimera, protac, targeted protein degradation, ubiquitin, proteasome, lysosome, autophagy

## Abstract

The induction of protein degradation in a highly selective and efficient way by means of druggable molecules is known as targeted protein degradation (TPD). TPD emerged in the literature as a revolutionary idea: a heterobifunctional chimera with the capacity of creating an interaction between a protein of interest (POI) and a E3 ubiquitin ligase will induce a process of events in the POI, including ubiquitination, targeting to the proteasome, proteolysis and functional silencing, acting as a sort of degradative knockdown. With this programmed protein degradation, toxic and disease-causing proteins could be depleted from cells with potentially effective low drug doses. The proof-of-principle validation of this hypothesis in many studies has made the TPD strategy become a new attractive paradigm for the development of therapies for the treatment of multiple unmet diseases. Indeed, since the initial protacs (Proteolysis targeting chimeras) were posited in the 2000s, the TPD field has expanded extraordinarily, developing innovative chemistry and exploiting multiple degradation approaches. In this article, we review the breakthroughs and recent novel concepts in this highly active discipline.

## 1. Introduction

Regulated protein degradation is performed mainly by the ubiquitin-proteasome system, the endocytic and the autophagy pathways, proteasomes and the lysosome being the two main proteolytic hubs in the cell. These highly complex systems account for the degradation and turnover of most of the proteins in the cell. Ubiquitin is a post-translational protein modifier involved in multiple cellular processes, representing the core of regulated protein degradation in eukaryotes and acting as a key signal in proteasomal, autophagic and endocytic pathways [1,2,3,4] (a scheme is shown in Figure 1). Protein ubiquitination generates a complex signaling code in tagged proteins, which includes modification with one single ubiquitin molecule (monoubiquitination) or with distinct types of ubiquitin chains (or polyubiquitin) conjugated to the substrate (polyubiquitination). The complexity of ubiquitin signaling relies on the fact that polyubiquitination produces topologically and functionally distinct polyubiquitin patterns, affecting a broad variety of regulatory aspects in the cell [5]. Different polyubiquitin types are produced by the modification of up to seven internal lysines in the ubiquitin sequence: K6, K11, K27, K29, K33, K48 and K63. Moreover, linear M1 polyubiquitin can be formed by linkages in methionine 1 of the ubiquitin sequence [5,6]. Each type of polyubiquitin chain is recognized by a different set of receptors, thus targeting polyubiquitinated cargos to distinct fates [7].

Ubiquitination is catalyzed by the sequential activity of ubiquitin-activating (E1), ubiquitin-conjugating (E2), and ubiquitin ligases (E3) enzymes, which promote the formation of ubiquitin-protein covalent conjugates with an isopeptide bond between the C-terminal glycine of ubiquitin (G77) and the lysine residue of the acceptor protein [2,8]. Deubiquitinating enzymes, also known as DUBs, catalyze the hydrolysis of the ubiquitin linkage, thus providing a level of regulation of the signal [9]. In the process of ubiquitination, ubiquitin is activated in an ATP-dependent manner by the E1, and then transferred to an E2, forming an E2-ubiquitin thioester adduct. Ubiquitin ligation to the accepting lysine requires an E3, which binds the E2 and recognizes the protein substrate, providing specificity [10,11,12]. Two evolutionarily and mechanistically distinct families of E3s have been described. The HECT E3s contain a cysteine active site and form a transient E3–ubiquitin complex in order to catalyze ubiquitin ligation [13,14]. The RING E3s, forming monomers, dimers and multimeric complexes, instead promote ubiquitination by placing the E2 and the protein substrate in the functional spatial context (Figure 2) [15,16,17,18,19]. The Cullin RING ligase (CRL) superfamily is a highly extended type of multimeric RING ligases, containing up to seven Cullin families, and some of their members are mentioned in the text (see Figure 2) [20,21,22,23,24].

The proteasome is a multi-protein complex consisting of a core particle (CP or 20S), a cylinder with proteolytic activity, and a regulatory particle (RP), which binds, unfolds and translocates polyubiquitinated proteins to the CP. The CP is composed of four stacked rings that contain alpha and beta subunits. The two inner rings formed by beta subunits contain three distinct proteolytic sites oriented towards the lumen of the cylinder. The two outer rings formed by alpha subunits seal the entrance to the active sites with the protruding N-terminal ends of subunits alpha3 and alpha7 [25]. The 20S structure confines the proteolytic activity in a space inaccessible to the cytosol. Thus, the opening of the entrance to the CP is required for substrates to reach the interior of the chamber. However, since the diameter of the inner tunnel of the CP is 28 Å [25], most proteins cannot enter in a native state. The RP is organized in two sub-particles: the base, which contains the substrate-unfolding ATPase ring and receptors (see Figure 1), and the lid, which contains the integral DUB activity that hydrolyzes the polyubiquitin linkage of substrates while they are translocated [26,27]. In the 26S (RP-CP) and 30S (two RPs and one CP) proteasomes, substrate recognition, mediated by the polyubiquitin interaction with the receptors, facilitates substrate tail engagement, proteasome conformational switches, deubiquitination, ATP-dependent substrate unfolding, and translocation, which culminates in substrate degradation in the CP [2,28,29,30]. In RP-less proteasomes, the presence of alternative activators promotes the opening of the gate and facilitates the entrance and degradation of mainly unstructured protein substrates [31,32].

On the other hand, in selective autophagy, the cargos interact with the LC3 family proteins, via ubiquitin-dependent or -independent mechanisms, thus providing selectivity and acting as a signal that initiates the process [3]. A key event in the process is the recruitment of the cargo, which is mediated by autophagy receptors [3,33] (Figure 1). Receptors are protein adaptors that contain ubiquitin-binding domains, which bind ubiquitinated proteins present in the cargo, and LC3-interacting regions, which bind with LC3 present in the nascent autophagosome. Distinct E3 ligases have been described to be involved in autophagy cargo ubiquitination [34,35,36] (see Figure 1). These specific interactions promote the formation of a selective autophagosome, which eventually will deliver its cargo to the lysosome [37].

In the last few decades, TPD has emerged as a novel therapeutic concept in which small-molecule ligands bind protein targets and redirect them to agents of the proteostatic machinery, in order to induce their degradation, thus acting as a degradative knockdown which inactivates the selected targets. In this context, the possibility of exploiting the cellular proteolytic systems for the design of innovative drugs has attracted the attention of researchers, since it offers the opportunity to overcome some of the limitations of classical pharmacology. Thus, as compared to classical target inhibition, TPD offers advantages that suggest that it is on the edge of a new generation of highly promising therapeutic compounds, not lacking in reasonable caveats, as discussed herein. For instance, a central paradigm in classical drug design is a mode of action mainly based on the inhibition achieved by the binding of compounds to the active or allosteric sites of the targets. This approach normally depends on the deep structural-functional characterization of targeted proteins and on the design of molecules that interfere with the activity of the target. In order to be efficient, inhibitory or activator drugs need to reach high concentrations in compartmentalized sites of the cell so as to ensure near full occupancy of the disease-related target [38]. As a consequence, the administered doses have to be high and long enough to make treatments efficient, with the subsequent off-target and toxic side-effects [38,39]. In the TPD approach, an outstanding trait is that compounds are re-engaged once the degradation is completed, participating in multiple degradation events, thus acting as catalysts or catalysis inducers. Therefore, target inactivation could be reached by substoichiometric drug/target ratios, which become even more favorable as the target is being depleted [40,41]. This aspect is addressed in multiple works in the field, in which degraders show activity in the nanomolar range, and is exemplified by a study on Tyrosine kinase receptor inhibition, where dedicated protacs outperform classical chemical inhibitors [42]. This feature opens the door to effective doses in the nanomolar range, with positive consequences for off-target and toxicity alleviation as compared to doses in the micromolar range.

An additional trait, in which TPD shows much higher potential than the conventional inhibitors, is that, in theory, any ligandable surface of the target can be used for ligand binding, thus it is not limited to active or allosteric sites, since the drug acts as a recruiter, not as an active-site modulator. This fact dramatically increases the potential proteome amenable to this approach, including proteins typically considered undruggable by classical pharmacology [43]. On the other hand, several limitations have to be considered when developing TPD molecules. Maybe the most prominent one is the difficulty in accomplishing the Lipinski rule of five [44], due to the usual large size and physicochemical properties of the compounds, mainly the heterobifunctional (chimeric) ones, as discussed below.

In this scenario, the effort of researchers are focused on creating novel TPD systems and molecules that give the opportunity to degrade a broad variety of disease-related protein targets with improved permeability, efficiency and scope, translating this approach to a vivid and exciting field. Of note, multiple excellent reviews have been released in the last few months giving deep information about the basis of TPD methodology [42,45,46,47,48,49,50]. In the present article, we will give a bird’s eye overview of the initial degraders, in order to specifically focus on the most novel and innovative concepts in this quickly changing discipline.

## 2. The Beginning of TPD

The initial hypothesis of induced protein degradation by an engineered molecule was posited in 2001 by Craig Crews and Ray Deshaies [51] (Table 1, which contains a list of the molecules reviewed herein). In that work, to determine whether a protein substrate (methionine aminopeptidase-2; MetAP-2) could artificially be targeted to the SCF complex containing the beta-TRCP F-box (SCF^bTRCP^ or CRL1b^TRCP^), a compound, which the authors named Protac-1 (Proteolysis-targeting chimeric molecule-1), that contained ligands for both the E3 and the substrate was synthesized. The IkBa phosphopeptide (IPP) was used as the SCF^bTRCP^ ligand, and ovalicin was used to recruit MetAP-2 (Figure 2A). Successfully, this new molecule was able to degrade MetAP-2. The importance of that discovery was probably underestimated at the time. One notable implication was that E3 ligases, by the formation of the proper protein-protein interaction, could easily accept neo-substrates, an aspect profusely corroborated in subsequent works, and which is, indeed, the mechanistic basis of the broad application of the protac approach. The large sizes of the first chimeric compounds (they contained a phosphopeptide with up to 18 amino acids) and their consequent low cell penetrance delayed their development and pharmacological application.

After this foundational idea, a considerable effort was made in order to generate druggable small-molecules with improved properties, based on the structural frame of the E3 ligand-linker POI ligand (POI is the protein of interest). In 2004, a novel protac system was developed, based on Von Hippel-Lindau E3 ligase (CRL2^VHL^) and on hypoxia inducible factor 1 (HIF1) α-derived peptide as a target [52]. The peptide-based protacs were successfully used in the degradation of Estrogen Receptor (ER) [53,54,55,56], the aryl hydrocarbon receptor [57,58], the hepatitis B virusX-protein [59], and also Tau [60], Akt [61] and Smad3 [62]. However, the molecules were still too large with challenging pharmacological applications. VHL protacs were remarkably improved with the replacement of the HIF1α peptide, used in the first generation, with a high-affinity small-molecule hydroxyproline ligand, critical for VHL binding [63]. This substitution improved the bio-orthogonality of the new compounds, one of the most important limitations of the approach. Small-molecule protacs were then developed, targeting the breakpoint cluster region—Abelson tyrosine kinase, BCR-ABL, by means of the inhibitors imatinib, bosutinib and dasatinib [64] (Figure 2B), the Bromodomain extra-terminal (BET) proteins [65] and the receptor tyrosine kinases [42].

In addition, a cell-permeable protac directed to another E3 ligase was developed in 2008. In this approach, the protac consisted of a non-steroidal androgen receptor ligand (SARM) and the MDM2 ligand known as nutlin, connected by a PEG-based linker (Figure 2C) [66]. In 2010, an additional protac type based on the cellular inhibitor of apoptosis protein 1 (cIAP1) E3 ligase, which is activated by methyl bestatin (MeBS), was defined. This generation of protacs, also known as “specific non-genetic inhibitor-of-apoptosis proteins (IAPs)-dependent protein erasers” (SNIPERs), recruit the homodimeric E3 cellular cIAP1 using the small-molecule ligand bestatin for POI degradation [67], and were initially developed to target the cellular retinoic acid-binding proteins (CRABP-I and II). To create this type of protacs, a hybrid molecule containing MeBS, all-trans retinoic acid and differently sized spacers was synthesized [67,68,69] (Figure 2D). SNIPERs have also been applied to successfully degrade ER alpha [70,71,72], the spindle regulatory protein transforming acidic coiled-coil-3 [73], BCR-ABL [74] and multiple HaloTag-fusion proteins [75]. SNIPER-based protein degradation exhibits the off-target binding of bestatin [76], and it further induces cIAP autoubiquitination and its subsequent degradation [77]. Even though the ligand has been optimized and cIAP1 autoubiquitination has been reduced, SNIPERs are still functional in the micromolar range [71].

Coetaneous to the development of chimeric heterobifunctional molecules, and complementary to them, a new concept appeared in the field: molecular glues (see also Section 3.4). The story of these potential therapeutic compounds is much more intricate, starting with a conspicuous drug repurposing. Thalidomide (α-(*N*-phthalimido)glutarimide) (Figure 2E), a drug prescribed during the 1950s to pregnant women against nausea, vomiting and anxiety, turned out to be highly teratogenic, and it was quickly withdrawn [78]. However, in the 1990s, the anti-inflammatory properties of thalidomide were shown, operating by inhibiting the release of tumor necrosis factor-alpha (TNF-α) from the peripheral blood monocytes (PBMCs) [79], as well as by enhancing the release of interleukin-2 (IL-2) and interferon-γ (IFN-γ) from activated T cells [80,81]. For these characteristics, thalidomide and its analogs were named immunomodulatory drugs (IMiDs). These drugs, being strictly prohibited to pregnant women in order to prevent embryopathy, can be administered with side-effects but no lethal effects in specific clinical cases [82]. Nonetheless, the definitive leap for thalidomide and its derivatives was made in proving their efficiency against multiple myeloma and other pathologies [83], and their mechanism of action, which defined the E3 ligase CRBN and the DNA-damage-binding protein 1 (DDB1) as the endogenous targets [84,85,86,87,88]. Remarkably, thalidomide does not act as an inhibitor, but as a CRBN binder by bridging protein partners, which, upon binding to the ligase complex, undergo ubiquitination and consequent degradation at the proteasome. The therapeutic applications of IMID compounds will be discussed in Section 3.4.

In the next sections, we will focus on the recent expansion of the protac methodology and its applications, and also on the development of novel concepts in TPD and chimera technology, which will undoubtedly have a strong impact in future pharmacology.

## 3. Modulating the Reactivity and Versatility of Proteolytic Chimeras

Once the concept and the applicability of proteolytic chimeras had been firmly established, efforts to modulate their reactivity by rational chemical modifications led to the development of new, more versatile chimeras. In the following sections, an account of some of the most representative evolved chimeras, whose mechanistic grounds are depicted in Figure 3, is presented.

### 3.1. Photocontrolled Protacs

Despite the accepted potential of protacs as selective, catalytic protein degraders, the possibility of off-target side effects cannot be overlooked. For this reason, the design of protacs endowed with an external, precise and tunable spatio-temporal control system has become an attractive option. In this context, several works using light to modulate the activity of protacs have recently been reported in the literature. Photocontrolled protacs are characterized by allowing the modulation of their active conformation by means of light of defined wavelengths [89,90]. One of the approaches is based on the design of a photoswitchable linker that responds to light by changing its geometry and, as a result, by altering the 3D disposition of both POI and E3 ligase linkers. One of most relevant approaches is based on the incorporation of an azobenzene photoswitch as part of the linker. Despite being a long-known phenomenon, the photoisomerization of azobenzene has found application in chemical biology quite recently [91]. The physical phenomenon that underlies the use of azobenzene as a photoswitch is the possibility of light-promoted *trans*-*cis* isomerization. The *trans* isomer is around 10 kcal mol^−1^ more stable than the *cis* one, which represents more than 99.99% predominance in the dark at equilibrium, according to the Boltzman distribution equation [92]. Interestingly, by irradiation at 340 nm (by π→π* excitation), a substantial amount of the *cis* isomer is produced, whereas the *trans* isomer can be regenerated again in the dark or by irradiation at 450 nm (n→π* excitation). Despite the fact that the change in the distance between the carbon atoms at the *para* position is around 3.5 Å, the molecular shape is dramatically altered upon irradiation, which justifies its use as a photoswitch for the spatio-temporal modification of protacs and other biomolecules [93].

There are several examples of the use of azobenzene as a linker component to render protacs that are activated as *Z*-isomers with blue-violet light (380 to 440 nm), while keeping inactive, as *E*-isomers, in the dark [94]. These protacs are usually referred to as PHOTACS [90]. A representative example is a PHOTAC addressed at the E3 ligase CRBN receptor to target the BET family of epigenetic readers BRD2-4 and FKBP12, and its fusion proteins. Thus, the BET inhibitor JQ1 was anchored to a thalidomide derivative addressed at CRBN through a linker containing an azobenzene moiety (Figure 4A). Studies on the cell viability of RS4;11 lymphoblast cells showed a significant difference in the activity of this protac upon irradiation with 390 nm light pulses for 72 h in comparison with incubations in the dark. These results were corroborated by a parallel light dependence degradation of the target BET proteins BRD2-4, as revealed by Western blot analysis [94].

One of the limitations of azobenzenes as photoswitches is the need for the biologically harmful UV light to induce the *E*→*Z* photoisomerization, via π→π* excitation at low wavelengths (around 390 nm). Moreover, an incomplete reverse *Z*→*E* photoisomerization, via n→π* excitation (at around 500 nm), is usually observed due to a partial overlap of this excitation band for both isomers. By introducing fluorine atoms at each of the *ortho* positions of the azobenzene moiety, together with a donating and an acceptor group (“push-pull”) or two acceptor groups (“pull-pull”) in each of the *para* positions, the resulting *ortho*-F_4_-azobenzenes show a strong bathochromic effect for the *E*→*Z* photoisomerization (around 530 nm), requiring less harmful, more penetrating wavelengths, which improves their biomedical applications [95,96]. In addition, due to the electronic effects of the substituents, the n→π* transition bands of the *E* and *Z* isomers can be separated enough to allow the selective and complete isomeric photoconversion, together with long photostationary states following the initial light excitation, which avoids the need for a continuous irradiation [97,98]. Photocontrolled protacs, which can be switched between *Z* and *E* isomers by irradiation at defined wavelengths, are referred to as photoprotacs [90]. Based on these premises, protac ARV-771 was modified into a *trans*-photoprotac for the generation of a photoswitchable BET degrader (Figure 4B). This photoprotac maintains the optimal distance between both warheads for the *trans* isomer and a roughly 3Å shorter distance for the inactive *cis* one [93].

An alternative way of controlling protac activity with light is by the design of photocaged protacs (pc-protacs) or opto-protacs [89]. The strategy of caging consists in the conjugation of a bioactive molecule with a protective group that results in a loss of function. A biomolecule is regarded as photocaged if the removal of the protecting group (or uncaging) is carried out by light [99]. Some examples of this application in protac design are found in the literature. For example, the addition of a photo-removable caging agent to a Brd4 degrader led to pc-protacs 1 and 2, with a potent degradation activity in cells after light irradiation [100]. Following the same concept, a pc-protac was also designed as a photocaged variant of an efficient Bruton tyrosine kinase (BTK) protein degrader [101]. In both cases, the 4,5-dimethoxy-2-nitrobenzyl (DMNB) group [99] was used as a photo trigger, since it can be efficiently cleaved upon irradiation at 365 nm. By adding the same caging group to the pomalidomide moiety targeting the E3 ligase CRBN, new opto-protacs addressed at dBET1 and dALK have been reported [102]. Similarly, this caging group has been incorporated into the VHL E3 ligase-recruiting ligand to afford the Brd4 pc-protac 3 shown in Figure 4C [103].

### 3.2. Covalent Protacs

As mentioned above, the degradation of biological targets by protacs is considered an event-driven process that takes place in a catalytic manner. For this reason, the design of most of the current protacs is based on the development of non-covalent interactions with both the POIs and the E3 ligase receptors. However, the uprising of covalent inhibitors as pharmacological tools and drugs [104,105] has not been unnoticed in the field of protacs, and some examples of covalent protacs have been reported in the literature. The first reported example was promising, although it was only tested in vitro [51]. In a recent study, a small collection of covalent BTK inhibitors was synthesized and, among them, protac 2 (Figure 5C) behaved as an effective degrader of the BTK protein [106]. A structural comparison with the inactive protac 3 [107] (Figure 5C) stresses the importance of protac design. Although addressed at different E3 ligases, both protacs incorporate an electrophilic Michael acceptor as a covalent warhead. However, the internal placement of this reactive moiety as part of the linker in protac 3 may be responsible, at least in part, for the lack of an effective binding with the target BTK. This is not the case in protac 2, where the Michael acceptor occupies a probably more accessible position.

Attempts to combine the advantages of an electrophilic warhead (leading to an irreversible covalent bond with the POI) with less reactive functional moieties, so as to restore the catalytic nature of protacs, led to the application of the concept of “reversible covalent inhibition” [108,109] to protac design. Reversible covalent inhibitors possess the potency and selectivity associated with the formation of a covalent bond, while being able to dissociate from the target protein once it is degraded. In this way, the issues associated with the potential immune response elicited by covalently modified proteins are practically abolished, as are the unpredictable long-term effects of such modifications, especially in the treatment of chronic diseases [110].

The rationale behind the structural modifications leading to a covalent reversible protac relies on the presence of a cyano group in the α position of the α,β-unsaturated Michael acceptor. The electron-withdrawing nature of the cyano group increases the electrophilicity of the warhead towards an alkylation reaction, while making more acidic the α carbonyl position for an efficient retro-Michael reaction leading to the release of the reactive protac. An example of this concept is found in the design of protacs against estrogen-related receptor α (ERRα). One of the members of the series (protac 1, Figure 5A) was able to degrade the ERR α protein by more than 80% at a low 30 nM concentration [111]. A very interesting work is the comparative study carried out with a series of non-covalent (NC), reversible-covalent (RC), and irreversible (IR) protacs against BTK, all of which were derived from the BTK binder ibrutinib [112] (Figure 5). The most potent reversible covalent protac (RC-3) exhibited enhanced selectivity toward BTK compared to the non-covalent (NC-1) and the irreversible covalent (IR-2) protacs used in the study. It should be noted, though, that the irreversible covalent protac IR-2 (Figure 5B) was very similar to protac-2 (see Figure 5C), already reported as an inefficient BTK degrader in cells [107]. In both cases, the connector used as a linker incorporates the reactive Michael acceptor’s moiety.

A strategy conceptually close to the concept of covalent protacs relies on the incorporation of a modified linker able to react with “HaloTag” fusion proteins. HaloTag is a modified bacterial dehalogenase that covalently reacts with hexyl chloride tags. HaloTag fusion proteins are currently used for the biorthogonal labeling of proteins in vivo, since plasmids for thousands of HaloTag-fused human gene proteins are commercially available [113]. Structurally, these protacs, directed to HaloTag fusion proteins, for which the generic term of “HaloProtacs” has been coined, are simpler than classical protacs, since they only require a binder for the ubiquitinating E3 ligase and a ω-chlorohexyl functionalized linker for covalent interaction with the HaloTag active site. As a proof of concept, a HaloProtac addressed at a HaloTag7-fused GFP has been designed and tested. In this work, a VHL ligand is differently functionalized with a series of ω-chlorohexyl-PEG linkers, differing in length and position on the VHL ligand (Figure 5C) [114]. The interest of this approach relies on their use as chemical probes to induce post-translational protein knockdown via the degradation of HaloTag7 fusion proteins, which can be routinely engineered by CRISPR/Cas9 genome editing technology [115] or by standard recombinant technologies derived from the commercial HaloTag7 plasmids [116].

In a more sophisticated approach, a VHL-derived HaloProtac recruiter has been used in combination with a HaloTag-fused high-affinity small polypeptide binder to develop “ligand-inducible affinity-directed protein missiles” (L-AdPROM). Thus, by producing a construct consisting of an anti-GFP nanobody (aGFP) conjugated to the HaloTag, the robust degradation of a GFP-tagged POI is observed only upon treatment of a variety of cells (A549, ARPE-19, HEK293, HEK293-FT, and U2OS) with the corresponding VHL-HaloProtac [117]. In this case, an antigen-stabilized aGFP mutant, only stable when bound to the antigen, was used to increase the specificity of the degradation machinery. This paper also illustrates the efficiency of camelid-derived nanobodies used as robust tools for selective target recognition, despite the requirement of rather elaborate POI-GFP and Halo-aGFP constructs [118].

### 3.3. In Cell Click-Based Protacs (CLIPTACs)

Small-molecule protacs are more promising than their peptide-based predecessors in terms of potency, metabolic stability and physicochemical properties. However, they still possess relatively large sizes (typically 700–1100 Da) and high polar surface areas (~200 Å^2^), which can limit their cellular uptake and compromise their bioavailability and pharmacokinetic properties, especially regarding their distribution across the central nervous system (CNS). Additionally, in order to achieve optimal protein degradation, a significant linker fine-tuning process is required, since a too-short linker may sterically prevent the formation of the POI:Protac:E3 ligase ternary complex, while an exceedingly long linker may fail to mediate the formation of the protein-protein interactions that are required for the ubiquitination reaction to take place. To overcome these limitations, an advanced protac technology named “in-cell click-formed proteolysis targeting chimeras” (CLIPTACs) has been developed. In the pioneering work of the Heightman group [119], a series of CRBN-based protacs, which are assembled intracellularly through a click-type biorthogonal inverse electron demand Diels-Alder reaction between two smaller precursors, was reported (Figure 6**) [118,119]**. The individual CLIPTAC precursors have smaller sizes and show a better cell permeability than the corresponding full protacs. Furthermore, when added sequentially to cells, the two clickable reaction partners were able to form a fully functional protac. Following this approach, the two key oncoproteins BRD4 and ERK1/2 were successfully targeted for ubiquitination by the CRL4^CRBN^ ligase complex for subsequent proteasomal degradation [119], as shown in Figure 6. However, no protein degradation was observed when the cells were treated with the full protacs obtained by the previous combination of the two clickable partners, suggesting that if the biorthogonal cycloaddition occurs outside the cell, the resulting cycloadduct cannot cross the cytoplasmic membrane [119]. Note that the CLIPTAC addressed at ERK1/2 was designed as a covalent protac, following the principles stated above.

### 3.4. Molecular Glues, Allosteric Modulators and Hydrophobic Tags

Molecular glues are small molecules that bind at the surface of E3 and/or target proteins, establishing contact interactions between both entities, leading ultimately to the ubiquitination of the target protein and to its subsequent proteasomal degradation [120]. In comparison with protacs, molecular glues are more attractive to drug development due to their smaller size and better drug-like properties. Molecular glues were first reported for plant hormones related to auxin and derivatives thereof. These small molecules are able to favor the interaction of the E3 ligase CRL1^TIR1^ and transcription factor targets, such as the Aux/IAA (Auxine/Indole-3-acetic acid) substrate. Interestingly, auxins increase the mutual affinity of both proteins by interacting in a small cavity at the protein-protein interface without inducing substantial conformational changes [121]. Recent examples of molecular glues are found in a series of anticancer aryl sulfonamides, collectively known as SPLAMs (splicing inhibitor sulfonamides). Among them, indisulam is used as an anticancer agent for its ability to promote the formation of a ternary complex with the E3 ligase receptor DCAF15 and the splicing factor RBM39, which is ubiquitinated and degraded by the proteasome. This splicing factor is overexpressed in some cancer cells lines, which become more sensitive to the cytotoxic effects of indisulam and related SPLAMs [122].

The fact that the discovery of molecular glues has been so far the result of serendipity has prompted researchers to define rational methods to discover novel molecules acting as glues [123]. Two recent papers have used distinct approaches, bioinformatics and molecular screening, to establish new molecular glue candidates, somehow converging in identifying and characterizing outstanding cyclin K degraders [124,125] (Figure 2). The Ebert lab bioinformatically analyzed the data of more than 4500 drugs tested against close to 600 cancer cell lines with mRNA levels of 500 E3 ligases, determining more than 67,000 correlations, and established a link between the Cul4 adaptor protein DDB1’s expression levels and the CDK inhibitor CR8’s [126] sensitivity. Subsequent functional and structural characterization defined a glue activity based on CR8 interactions with the ATP-binding pocket of CDK12 and the BPC domain of DDB1, the latter one being mediated by a phenylpyridine group that stands out from the CDK12 pocket to bind specific motifs of the DDB1 partner. The binding affinity and the tightness of the formed ternary complex directly correlate with the efficiency in ubiquitination, and thus with degrader glue capacity, as shown by testing distinct CDK family members, CDK12 mutants and CDK inhibitors similar to CR8 [125]. Remarkably, in another work, Qi’s group, by using a different approach, defined a distinct DDB1-CDK12 molecular glue, named HQ461, exhibiting some similarities to CR8 [124]. Lv et al performed a high-throughput screening of small molecules showing NRF2 inhibitory activity. They obtained more than 500 hits, with HQ461 among them. Even though the criteria to select HQ461 and no other hits is not mentioned in the work, the characterization of this compound’s activity showed its dependence on the CRL4^DDB1^ E3 ubiquitin ligase complex. Moreover, by an exome sequencing approach, they discovered that cells incorporating mutations in the CDK12 gene (at the position Glycine 731) acquired HQ461 resistance. HQ461, like CR8, induced the ubiquitination and depletion of cyclin K (or CCNK) in cells, acting as a molecular glue. Interestingly, HQ461 and CR8 exhibit structural similarities, especially in the pyridine end, involved in CDK12 recruitment. Therefore, these contributions demonstrate that a rational approach may lead to the identification of novel, so far very scarce but highly valuable, molecular glues, which can be further used as ligands or scaffolds of bimodal chimeras.

One type of small molecules that share some of the properties of molecular glues are the allosteric modulators, which can also bind an E3 receptor. However, unlike molecular glues, allosteric modulators may promote non-native interactions of the target ligase, leading to the ubiquitination of other proteins as neo-substrates. This is the mode of action of the immunomodulatory drugs (IMiDs) related to thalidomide, mediating the neo-substrate interactions of the E3 ligase receptor CRL4^CRBN^ with the transcription factors IKZF1 (Ikaros) and IKZF3 (Aiolos), which justifies the clinical application of IMiDs in multiple myeloma [88,127,128].

A small molecule ligand can also alter the conformation of its own target protein to promote its degradation. This interesting mechanism of action has recently been reported to account for the activity of the tuberculostatic prodrug pyrazinamide. Its active metabolite (pyrazinoic acid, POA, Figure 7A) is known to inhibit the biosynthesis of coenzyme A in *Mycobacterium tuberculosis* by binding to the aspartate decarboxylase PanD [129,130]. However, the fact that POA behaved as a weak PanD inhibitor at high concentrations was indicative of an alternative mechanism of action. In a recent work [131], POA was shown to stimulate PanD degradation via caseinolytic protease P (ClpP), a serine protease playing an important role in the proteostasis of eukaryotic organelles and prokaryotic cells [132]. This “event-driven” mechanism had a precedent in a series of selective estrogen receptor down-regulators (SERDs), a subclass of antiestrogens characterized by inhibiting estrogen binding to its receptor and by inducing a proteasome-dependent receptor degradation. This is the case of compounds ICI164,384, [133] RU58,668, and ICI182,780 (Faslodex^®^, AstraZeneca, Cambridge, UK), approved for the treatment of hormone-receptor-positive breast cancer [134] (Figure 7A).

The SERDs shown in Figure 7B are estrogens that have been modified with a long hydrophobic tag. The inclusion of a hydrophobic tag into a ligand may trigger, upon formation of the protein–ligand complex, a process called “unfolded protein response” (UPR), by which the exposure of hydrophobic residues to the solvent may be recognized by molecular chaperones as the signal of a misfolded protein. These chaperones can either rescue the misfolded protein or promote its degradation by the proteasome when refolding fails [135,136]. Despite the fact that the above SERDs were not initially designed as hydrophobic tags, this alternative mode of action cannot be ruled out. In general, adamantyl and Boc_3_Arg are the most commonly used hydrophobic tags to trigger protein degradation when attached to specific ligands. An adamantyl tag has been incorporated into several androgen receptor antagonists to generate a novel class of s SARDs of use in androgen-dependent cancer cell lines [137] (Figure 7B). The pseudokinase Her3 has also been targeted with the adamantane ligand TX2-121-1 [138], whereas examples of the application of the Boc_3_Arg hydrophobic tag have been reported in trimethoprim to target dihydrofolate reductase (thus opening up new ways to design antibacterial drugs) [139], as well as in modified diuretics, such as the ethacrynic acid derivative EA-B_3_A (Figure 7B), and other ligands designed to target glutathione-S-transferases [140,141].

### 3.5. Ubiquitin-Independent Protacs

Despite the fact that the above Boc_3_Arg tags have been initially regarded as hydrophobic tags triggering the UPR cellular machinery, the mechanism of induced degradation by Boc_3_Arg seems to differ from the classical adamantyl hydrophobic tags. In this context, a recent study showed that Boc_3_Arg-modified ligands stabilize and localize the target protein to the 20S proteasome, without requiring ubiquitination (see Figure 3). Likewise, purified 20S proteasome is apparently enough to degrade target proteins in the presence of their respective Boc_3_Arg-linked recognition ligands [141]. An example of Boc_3_Arg-ligand application is shown in protacs targeting Proprotein convertase subtilisin-like/kexin type 9 (PCSK9), a serine protease involved in the protein-protein interaction with the low-density lipoprotein (LDL) receptor. Blocking this protein-protein interaction prevents LDL receptor degradation and decreases LDL cholesterol levels, which makes PCSK9 a potential anti-atherosclerosis target. Merck has developed a set of PCSK9 small-molecule binders in order to create specific protacs. The best hit was found to be an allosteric interactor, which was optimized to generate a binder exposing a moiety suitable for functionalization. Of note, protacs based on E3 ligases did not induce degradation; instead, the Boc_3_Arg ligand could induce a remarkable decrease in PCSK9 endogenous levels, even though complete target depletion was not reached [142].

Another example of an ubiquitin-free strategy for targeted protein degradation is found in a recent patent, in which bifunctional molecules comprising an Usp14 binding partner were linked to a target protein binding partner [143]. Usp14 is a stoichiometric subunit associated to the 19S regulatory moiety of the proteasome. It plays multiple functions, such as protein substrate deubiquitination, blocking of the regulatory subunit Rpn11, and slowing down protein degradation. Interestingly, the inhibition of Usp14 enhances the proteasome’s proteolytic activity [144,145]. This mode of targeting could be ubiquitin-independent because the substrate is presented to the proteasome by its positioning near the Usp14-Rpn1 region through the action of the ligand, without the requirement of previous ubiquitination. Nevertheless, the putative action of proteasome-associated E3s [146,147] should not be excluded. Further characterization is required to shed light on this relevant mechanistic aspect.

### 3.6. Antibody-Chimeric Degrader Conjugates

Pillow et al [148] identified a novel chimeric degrader molecule based on the VHL binding moiety of previous protacs (MZ1 and ARV771) and a pyrrolopyridone-derived BET inhibitor. The novel molecule achieved complete degradation of BRD4 with a DC50 value of 0.03 nM as determined by quantitative immunofluorescence on the EOL-1 AML cell line. However, the molecule showed poor drug metabolism and pharmacokinetics results, in accordance with its physicochemical characteristics. The authors then explored the possibility of using antibodies and drug conjugation technology, originally intended to deliver cytotoxic payloads to the cell, to deliver protacs [149]. They functionalized the chimeric degrader via the introduction of a small disulfide-containing linker via a carbonate moiety attached to the hydroxy-proline of the protac. This methanethiosulfonyl (MTS)-containing moiety can then react with the engineered Cys residues in CLL1 (C-type lectin-like molecule-1)-engineered antibodies. CLL1 is overexpressed in AML-related cells [150] and has been validated as an antigen for the delivery of antibody-drug conjugates (ADCs) to acute AML cell lines [149]. These conjugates are expected to release the payload upon internalization and disulfide reduction in the lysosome.

When administered intravenously in mice with HL-60 AML xenografts, the ADC achieved dose-dependent tumor growth inhibition, contrary to either the unconjugated form of the chimeric degrader or the antibody alone. Additionally, the ADC was well tolerated and remained stable at the in vivo efficacy dose of 5 mg/kg. These data encourage the idea of using antibody conjugation to overcome the poor bioavailability often associated with chimeric degraders [151,152].

In a continuation article, Dragovich et al. [153] described the construction of several degrader-antibody conjugates. Again, some of these have poor solubility due to their physicochemical properties and need to be engineered or conjugated to antibodies in order to be functional in vivo. Initially, they centered their efforts on producing ERα degraders based on endoxifen, a tamoxifen metabolite, as warhead, bound via a linker to an XIAP or VHL-interacting moieties. However, conforming to the somewhat hydrophobic nature of the XIAP-based degraders, the produced molecules faced solubilization problems that were aggravated when conjugated with an antibody. Despite efforts to change the chemistry of the drug-antibody linker, the initial compound was considered unsuitable for in vivo studies because the compound faced in vivo biotransformation.

The authors then changed their approach and switched to an ER degrader based on VHL E3, conjugated to antibodies again via MTS moieties, resulting in disulfide-based links, but with slightly different chemistry, as a carbonate group was used to connect the disulfide linker. These molecules were protected from unwanted biotransformation by the incorporation of a methyl group adjacent to the aforementioned carbonate group.

In this second attempt, the authors achieved an ADC with ERα degradation activity in vivo and with demonstrated selectivity for HER2+ cells. Finally, the authors explored an alternative conjugation method for the Endoxifen–VHL chimera using a pyrophosphate di-ester moiety to connect the protac to the maleimide used in the bioconjugation, with preliminary results suggesting specific and efficient intracellular release of the payload.

Maneiro et al. [154] developed a trastuzumab-BRD4-degrading chimera ADC to promote BRD4 degradation specifically in HER2+ cancer cells. Conjugation is achieved by rebridging the interchain disulfide bonds of trastuzumab with next-generation maleimides (NGMs), achieving a drug:antibody ratio (DAR) of 4, protecting the conjugation from early biotransformation and ensuring that the protac is only released after internalization.

After 4 h treatment with 100 nM ADC, BRD4 was fully degraded only in HER2+ cells, while remaining unaffected in HER2− cells. Additionally, the authors fluorescently labeled the ADC and were able to follow its trafficking from cell surface to lysosomes, where the protac molecule must undergo cleavage from the antibody and activation followed by transport to the nucleus to achieve BRD4 degradation.

Clift et al [155,156] developed a method, named Trim-away, for rapidly depleting the cells of a POI recognized by a specific antibody. Firstly, the antibody is delivered by microinjection or electroporation. Then, the FC in the antibody is recognized by the E3 ubiquitin-protein ligase TRIM21. This complex is a cytosolic receptor that participates in humoral immunity by ubiquitinating intracellular pathogens and marking them for proteasome degradation [157], while also having a role in fighting pathological protein aggregates [158]. In cell lines where TRIM21 is not sufficiently expressed, the recombinant complex can be co-electropored with the antibody.

In their study, the authors tested the system against a variety of substrates, such as cytosolic-free GFP, GFP fused to the histone H2B, membrane-anchored GFP or GFP fused to a nuclear localization signal. Trim-away was very efficient in degrading these substrates in a cytosolic context, but due to the size of the antibody part of the system, it could not interact with proteins residing in an intact nucleus. The authors then expanded the capabilities of the system by using a smaller FC—nanobody fusion, allowing for the degradation of nuclear proteins.

In a similar way, Ibrahim et al. [159] devised a method based on the specific interaction of an antibody fused to the RING domain of the ubiquitin E3 ligase RNF4 (ARMeD, antibody RING-mediated destruction). The objective of this work was to engineer a single-component system that could be easily produced and used as a reagent to induce POI degradation. They worked with the nanobody fused to either a RING domain or two RING moieties, making for a constitutively activated ligase. The construct maintained the NLS, which allowed it to interact with nuclear substrates.

The system was tested in mammalian cells harboring a dox inducible, anti-GFP nanobody construct. Upon induction, the levels of the reporter protein ADP ribose glycohydrolase (YFP-PARG) lowered steadily as the construct was expressed, with 19-fold reductions over the course of 24 h and a half-life of 7 h. The construct also proved active against an especially stable substrate, YFP-fused PML (promyelocytic leukemia) protein, present in nuclear bodies. In a second trial the authors raised a nanobody against unmodified NEDD8-Specific Protease NEDP1, effectively bringing it to undetectable levels by 12 h.

Finally, the authors delivered pg quantities of the construct to cells, achieving 85% of degradation within 10 min of electroporation. These effects, however, lasted for 6 h. In order to expand the depletion effect, the protein had to be co-electroporated with codifying mRNA, thus effectively extending the effects over 24 h.

## 4. Expanding the (Sub)Cellular Landscape of Targetable Proteins (Factors)

In parallel with the extraordinary innovation in protac design from the molecular and chemical standpoint, overviewed in previous sections, remarkable breakthroughs in target localization and proteolytic pathways have been achieved. Some of the most representative are addressed below.

### 4.1. Protacs for Solute Carrier Proteins (SLC-Protacs)

How universal is the use of E3-based protacs in terms of target subcellular localization? Is it mainly restricted to accessible cytosolic proteins and to membrane proteins with one or two transmembrane motifs? A recent work from Bensimon and collaborators sheds light on this relevant point. They developed protacs against a set of SLC proteins (with multiple transmembrane motifs and diverse localizations) conceived as CRL4^CRBN^ hijackers, and uncovered a potential field of targetable proteins [40]. They showed that multi-span transmembrane SLC proteins can be very efficiently degraded by CRBN-mediated degradation from multiple subcellular localizations, including plasma membrane, ER, Golgi (see Figure 1), and the outer mitochondrial membrane (albeit less efficiently). Moreover, they discovered that plasma membrane proteins can be degraded by TPD from any transient localization along the process of synthesis and maturation until they reach their final status. The only requirement is the cytosolic orientation of the ligand-interacting domain. They initially used the dTAG system, which appends a mutated FK506-binding protein (FKBP12) to the tagged POI, utilized as the tag that enables the phthalimide-mediated degradation of the POI by a variety of chimeric degrader molecules (e.g., dTAG7/dTAG13) that simultaneously bind to the dTAG and the CRL4^CRBN^ E3 ligase [160]. Moreover, they developed a protac for one of the best-behaved hits, SLC9A1 (or NHE1), a cancer-related Na^+^/H^+^ ion transporter with an important role in cytoplasmic and microenvironmental pH regulation [161]. Of note, the novel protac, d9A-2 (Figure 2F), could degrade SLC9A1 from leukemic cells, in a nanomolar 8 h treatment, exhibiting a really promising drug profile. Therefore, it could be concluded that the first generation of protacs targeting multi-span plasma membrane protein (SLC9A1 contains 12 transmembrane domains) has been successful, thus foreseeing multiple applications and highlighting the versatility of the CRL4^CRBN^ ligase-protac approach.

### 4.2. Lysosome-Targeting Chimeras (Lytacs) for Endocytically Internalized Targets

All this notwithstanding, when considering targeted protein degradation to tackle human diseases, it becomes clear that the actual protein substrates intended to be removed from the cell are usually not accessible to the proteasome, and therefore, the conventional protac strategy is not viable. This is the case of certain compartmentalized proteins, extracellular factors, proteinaceous aggregates, and proteasomal-refractory polypeptides. In these cases, recent contributions have provided multiple possibilities in terms of pathway exploitation and molecule innovation.

Bertozzi and collaborators, in a remarkable contribution, presented Lytacs, chimeras capable of targeting proteins for destruction in the lysosome [162]. To do so, they developed bimodular molecules with ligands that bind both a cell surface lysosome-targeting receptor and a protein targeted for degradation, which, in this case, is not an intracellular or cytosolic protein but an extracellular or a plasma membrane protein. Lytacs induce the internalization and the lysosomal degradation of the target. The authors focus their strategy on a prototypical lysosome-targeting receptor, the cation-independent mannose-6-phosphate receptor (CI-M6PR, or IGF2R, Insulin-like growth factor 2 receptor), which plays a role in transporting proteins modified with N-glycans, capped with mannose-6-phosphate (M6P), to lysosomes [163]. The CI-M6PR receptor shuttles the cargo to the pre-lysosomal compartments, where low pH values induce receptor-cargo dissociation. The targeted protein is delivered to the lysosome and CI-M6PR is then recycled to the plasma membrane.

In a brilliant design, Banik et al. take advantage of this endogenous mechanism to develop a degradative tool with promising high-efficiency applications in cancer, neurodegeneration and multiple additional diseases. For example, versatile ligands adopted by Lytac technology could conceive chimeras capable of inducing the degradation of extracellular aggregation-prone proteins involved in a degenerative disease, viral particles on their way to infect cells, or plasma membrane receptors acting as signal transducers in oncogenic processes, just to mention potentially high-impact applications. They indeed observed that the conjugation of a poly(M6Pn)-bearing glycopolypeptide to an antibody (Figure 8A) successfully endowed the antibody with the capacity to traffic extracellular factors for destruction into the lysosome. They non-specifically labeled lysine residues on a polyclonal anti-mouse IgG with bicyclononyne-*N*-hydroxysuccinimide (BCN-NHS), and then conjugated the antibody to azide-terminated M6Pn glycopolypeptides via copper-free strain-promoted azide-alkyne cycloaddition, generating the Lytac Ab-1. This innovative molecule was able to induce the transport of Alexa FluorF-488-labeled mouse IgG to the lysosome, which opened up the possibility of generating a tripartite interaction of (i) Ab-1, (ii) a primary IgG and (iii) its antigen, and inducing the terminal traffic of captured antigens towards the lysosome (see Figure 1). This important point was corroborated by using mCherry and anti-mCherry IgGs, and further validated with apolipoprotein E4 (ApoE4), a factor involved in neurodegenerative diseases. The presence of Ab-1, Anti-ApoE primary antibody and ApoE-647 induced a 13-fold increase in the target uptake and the detection of the fluorescent signal in the lysosome during the 24 h period of continuous internalization. This result confirmed the feasibility of the Lytac approach in targeting a clinically relevant factor for lysosomal degradation.

Remarkably, Lytacs were shown to be efficient in the degradation of plasma membrane integral proteins, as well. The method was carefully validated with the cancer-related EGF receptor (EGFR) and with cetuximab, used as an EGFR-capturing antibody and as a control of the assay. They observed more than 70% degradation of EGFR in HeLa cells, in the presence of cetuximab functionalized with M6Pn glycopolypeptides. This degradation was dependent on CI-M6PR, since knocking down the encoding gene, *IGF2R*, completely blocked EGFR degradation. Importantly, this innovation generated an optimized version of cetuximab, which, functionalized as a Lytac, was able to catalyze the depletion of EGFR, offering notable advantages with respect to cetuximab per se. One of them is that all the scaffolding and kinase-independent roles of EGFR are totally impaired upon Lytac treatment, but not in cetuximab-treated cells (e.g., the EGFR auto-phosphorylation effects induced in cetuximab-inhibited EGFR are not observed in cells treated with functionalized cetuximab). The Lytac method was also efficacious in degrading additional cancer-related membrane proteins, such as CD71 and PD-L1. The only caveat to this first set of observations is that when the clearance of functionalized and non-functionalized antibodies (cetuximab) was compared in an animal model, a decay of M6PN-cetuximab levels was observed only during the first 6 h, whereas in the 6–72 h period, a modest clearance was observed. Understanding and optimizing this double-phase kinetics will be important in order to implement these outstanding new tools at the clinical level.

Overall, Bertozzi and collaborators defined a generation of compounds with a very wide scope of therapeutic and functional applications. Some of the attractive innovations of Lytacs is that they recruit targets dwelling in the extracellular space and the cellular membrane, and that the recognition of the target can be achieved by not only ligand-surface interactions, but also by antibody-antigen engagement. Therefore, this methodology could be applied to degrade plasma membrane receptors which act upstream of signal transduction pathways, channels and transporters, with important effects in multiple pathologies, for example, in cancer proliferation. Furthermore, the possibility of acting on extracellular proteins provides a rationale for the design of degraders for viral factors and for proteins potentially toxic in neurodiseases.

### 4.3. Autophagy Targeting Chimeras

#### 4.3.1. Autacs

In a recent article, the Arimoto group showed a novel type of targeting chimeras focused on selective autophagy. They developed a sort of autophagy-protacs, since these chimeras induce the polyubiquitination and subsequent recruitment of selective autophagy factors [164]. They base their development on the capacity of S-guanylation to recruit the selective autophagy machinery into a POI, promoting K63-linked polyubiquitination, recognition by SQSTM1/p62, LC3 binding and autophagosome formation. By engineering an EGFP HaloTag labeling system associated with cGMP, they could induce autophagy on (EGFP-HT)-(HTL-cGMP). However, this tagging system appeared to be extremely slow and produced side effects. Thus, in order to find a more efficient labeling method, they tested guanine derivatives and discovered that p-fluorobenzyl guanine (FBnG) could mimic S-guanylation and recapitulate autophagic degradation with improved orthogonality. When HeLa cells expressing EGFP-HaloTag were treated with FBnG-HTL, they observed the production of EGFP autophagic dots, and colocalization with LC3, p62/SQSTM-1 and with K63-linked polyubiquitin, resulting in a 70% EGFP depletion. Moreover, this experimental proof-of-principle allowed the authors to characterize the specificity of the FBnG-targeting compound, observing dependence on Atg5 and no dependence on proteasome activity.

Then, they created the first generation of Autacs on the basis of three elements: FBnG, a polyethylene glycol linker, and a ligand for a POI. With this strategy, they promoted the selective autophagy and efficient degradation of (i) MetAP2 with a fumagillol-based Autac, (ii) FKBP12 with the FKBP synthetic ligand (FSL), and (iii) the Bromodomain of BRD4 using JQ1 acid as a warhead. The nuclear localization of BRD4 made the Autac approach more challenging in this case, since autophagy is mainly cytosolic. The synchronization of cells allowed them to determine a phase-dependent degradation of the target, promoted by the Autac, which took place during the G2-to-G1 transition, when the nuclear envelope was destroyed in order to allow the progress of mitotic phases. The absence of nuclear membrane made the interaction of LC3 with nuclear proteins possible.

Notably, the Arimoto group were also able to induce mitophagy using FBnG guanine derivatives. To test this point, they expressed in cells an EGFP-HT-Omp25 fusion protein, thus labeling the outer membrane of mitochondria (Omp25 is an OMM integral protein) with a fluorescence-trackable and FBnG-interacting protein. However, the S-guanylation signal per se was not sufficient to promote autophagy, and only when mitochondrial fragmentation was induced, by the silencing of dynamin-like protein Opa1 or by the depolarizing agent carbonyl cyanide m-chlorophenylhydrazone (CCCP), did fragmented mitochondria become responsive to FBnG-HTL treatments. Furthermore, authors developed a mitochondria-binding compound (Autac 4) using a 2-Phenylindole-3-glyoxyamide-PEG-FBnG configuration (Figure 8B), also functional in the context of mitochondrial fragmentation.

The fact that treatment with CCCP was required for Autac4 mitochondrial degradation gave the opportunity to evaluate whether the turnover and new biogenesis of mitochondrial pools could ameliorate the toxic effects of depolarization. Indeed, the partial restoration of membrane potential was observed, monitored by decreases in cytochrome c release, and the activation of caspases and apoptosis, showing a promising protection effect of the small-molecule-induced degradation of damaged mitochondria, followed by regeneration of the organelle. This important healing effect of Autac4 has therapeutic applications in degenerative pathologies. As the authors point out, mitochondrial dysfunction is a key alteration in Down Syndrome (DS) etiology [165]. Therefore, they tested Autac4 in DS cells and observed encouraging improvements in membrane potential, mitochondria biogenesis and ATP metabolism. Further research is required to determine the potential use of Autac4 for drug development.

#### 4.3.2. Autophagosome Tethering Compounds (Attecs)

An additional and complementary approach to hijacking autophagy for TPD was developed recently by Lu’s group [166]. In this work, authors screened a glass-immobilized small molecule microarray for compounds able to simultaneously and specifically bind a mutant Huntingtin allele (mHTT) containing a polyglutamine repeat and LC3B. They found up to four compounds, named 10O5, 8F20, AN1 and AN2 (Figure 8C), which behaved as mHTT-LC3B linkers that could induce the turnover of mHTT in an autophagy-dependent manner, causing a substantial lowering of the levels of the toxic allele. This approach is quite interesting, since its mechanism of action does not require either ubiquitination or autophagy receptors, directing the cargo straight away to LC3. Authors tested the compounds in several cellular models, including cells from patients with Huntington disease, with consistent decreases in mHTT. When injected intraperitoneally in mHTT knock-in mice, AN2 and 10O5 were able to cross the blood-brain barrier, causing a significant decrease in mHTT and rescuing some of the causative phenotypes. These promising compounds exhibited selectivity towards long poly-Q proteins, with a threshold in the 25–38 glutamine range, found not only in mHTT, but also in other poly-Q proteins causing neurodisease, such as mutated Ataxin 3 (ATXN3), as demonstrated by the authors.

## 5. Miscellaneous Protacs

### 5.1. “Bioprotacs”

Bioprotacs are engineered fusion proteins that consist of a target binding domain and an E3 ligase, an arrangement that results in the specific degradation of the therapeutic target [167].

### 5.2. Conformationally Restricted Protacs

Very recently, a macrocyclic protac has been designed as a conformationally restricted analog of the BET degrader MZ1 [168]. As it is common in classical drug design, the use of this conformationally restricted analog leads to a more selective degrader with a cellular activity comparable to that of the parent flexible protac.

### 5.3. N-Degron Pathway-Based Protacs

A novel E3-targeting system has been proposed recently. This approach briefly leverages the N-end rule pathway, a system in which the amino acid residues occupying the N-terminal position of a protein are subjected to processes such as deamidation and arginylation, and are eventually recognized by the UBR1 E3 ligase and targeted for degradation in the proteasome [12]. In this contribution, Lee et al. [169] utilized the tetrapeptide N-terminal degron LRAA as a UBR1 binder, to build a chimera in which the target ligand is YL2, a helical motif that binds the steroid receptor coactivator-1 (SRC1), mimicking the specific interaction between the Signal Transducer and Activator of Transcription (STAT6) and SRC1. SRC1 regulates the expression of multiple additional transcription factors, and its levels correlate with metastasis and recurrence [170]. The authors tested the N-degron LRAA-YL2 protac in several cancer cell lines and observed efficient degradation and a significantly reduced invasion capacity of cells treated with the novel degrader. This is a promising strategy that may offer an opportune alternative to classical E3 protacs, in cell types and conditions in which ligase levels are limited, as authors point out.

## 6. Concluding Remarks: An Exciting Third Generation of Protein-Degrading Chimeras

Chimeras inducing proteolysis broke out as low-profile molecules due to the multiple limitations they suffered from large size, low cell penetrance and metabolic processing. This was the first generation of protacs, which provided by the 2000s a valuable proof-of-principle of a novel way to silence a desired protein but which exhibited poor chances of survival in the competitive jungle of rationally designed drugs. Not much later, in the mid-2010s, notorious breakthroughs in small-molecules capable of exquisitely selecting E3 ligases boosted the projection of TPD as a likely approach to create new drugs with improved capacity of proteolytically inactivated disease-related proteins. This second generation is indeed currently providing new molecules that, due to their potency and catalysis-based mechanism of action, may show advantages over standard occupancy-based drugs. New protacs show activity in the nanomolar range, deplete the disease-causing proteins, and tackle typically non-druggable targets. Moreover, they may be more efficient against both low-concentration and acutely overproduced toxic factors found not only in cancer, but also in neurodegeneration and multiple diseases.

Furthermore, while the expansive wave of second generation protacs is still propagating, a third generation of chimeras has overlapped. This new generation goes beyond the exploitation of the ubiquitin-proteasome system, and the target scope is not restricted to available cytosolic proteins. Recent papers report on molecules that target intracellular factors in autophagy or extracellular internalized proteins for lysosomal degradation. These include compounds that induce the engulfment of specific cargos, including mitochondria, by the autophagosome to promote their targeted destruction or antibody-based degraders, among others. Moreover, the development of protacs and other chimeras has paralleled that of the available chemical toolbox. For example, light can be used for the precise activation or deactivation of protacs with exquisite selectivity and precision. Additionally, although still in its infancy, the repertoire of already available bio-orthogonal reactions has been rationally applied to the design of smaller, more “drug-like” protac precursors, able to react intracellularly to render the active chimera.

From a clinical perspective, a non-negligible number of therapeutically relevant proteins have been targeted using the protac technology. Despite the fact that only two compounds are currently in clinical trials for some varieties of resistant breast and prostate cancers [171], this number is expected to rise in the forthcoming years.

Altogether, a novel biomedical discipline emerges, sustained by a constantly expanding methodology and applicability with no boundaries, in which high-efficiency drugs will not only deplete specific protein targets, but will also induce all kinds of regulatory events to multiple types of targets, by means of completely new molecules, such as chimeras targeting to ribonucleases (Ribotacs) [172] or to phosphatases (PhoRCs) [173], and much more, in a still unthinkable tailored pharmacology.

**Table 1 molecules-25-05956-t001:** Degrader molecules discussed in this review.

Type	Targeted Protein	Degradation Factor	Therapeutic Potential *	Reference
Protac	Multi-span transmembrane SLC proteins	CRBN	Oncology	[40]
Protac	ErbB1/HER1 (EGFR)	VHL	Oncology	[42]
Protac	MetAP-2	SCF complex (SCF^bTRCP^)	Research use, first protac	[51]
Protac	Estrogen-related receptor alpha and RIPK2	VHL	Immunology: Blau syndrome (RIPK2). Oncology: breast cancer(ERRa), early-onset sarcoidosis (RIPK2)	[63]
Protac	BCR-ABL	CRBN or VHL	Oncology: Chronic myelogenous leukemia	[64]
Protac	BET proteins	VHL	Oncology: castration-resistant prostate cancer. AR-related cancers	[65]
Protac	Androgen Receptor	MDM2	Oncology: AR-related	[66]
Protac	Retinoic acid, estrogen and androgen receptors	cIAP1	Oncology: CRABP-II-targeting; oncology: therapy for controlling tumor metastasis	[67]
Allosteric modulator	Ikaros and Aiolos and casein kinase 1α	CRBN	Oncology: Multiple myeloma, B cell malignancies	[88]
Photoactivated protac	BRD2-4 and FKBP12	CRBN	Oncology, precision medicine	[94]
Photoactivated protac	BRD4, BTK	CRBN	Oncology	[100]
Photoactivated Protac	Bruton tyrosine kinase (BTK)	CRBN	Oncology: Precision medicine	[101]
Photoactivated protac	BRD2/3 and Anaplastic lymphoma kinase (ALK)	CRBN	Oncology, precision medicine	[102]
Photoactivated protac	BRD4	VHL	Oncology, prescision medicine	[103]
Covalent protac	BTK and BLK	CRBN or VHL	Oncology	[106]
Covalent protac	ERRα	VHL	Metabolic disorders: Type II diabetes. Oncology: Her2+ and triple-negative breast tumors	[111]
Covalent protac	BTK and BLK	CRBN	Oncology: chronic lymphocytic leukemia	[112]
Covalent protac	Platform for Halo-tagged proteins.	VHL	Research use	[115]
Covalent protac	Platform for GFP-tagged proteins	VHL	Research use	[117]
Click protac	BRD4 and ERK1/2	CRBN	Oncology	[119]
Molecular glue	RBM39	E3 ligase receptor DCAF15	Oncology	[122]
Molecular glue	RBM39, cyclinK	CRBN	Oncology	[123]
Molecular glue	DDB1-CDK12 molecular glue	CRBN	Oncology	[124]
Molecular glue	CDK12-cyclin K	CRBN	Oncology	[125]
Allosteric modulator	Aspartate decarboxylase PanD	E3-independent	Research use, anti-microbial agents	[131]
Ubiquitin-independent degrader	Androgen receptor (AR) (F876L)	ND (indirectly, CHIP)	Oncology: prostate cancer	[137]
Ubiquitin-independent degrader	Her3 (ErbB3)	E3-independent (Hsp70 and Hsp90 chaperone mediated)	Oncology: breast cancer	[138]
Ubiquitin-independent degrader	eDHFR and GST- α1 GST-π	E3-independent	Research use	[140]
Ubiquitin-independent degrader	GST proteins, eDHFR	E3-independent	Oncology	[141]
Ubiquitin-independent degrader	Proprotein convertase substilisin-like/kexin type 9(PCSK9)	E3-independent (20S targeting)	Vascular disease	[142]
Ubiquitin-independent degrader	BRD2/4	E3-independent	Oncology	[143]
Antibody-conjugated protac	BRD4	VHL	Oncology, precision medicine	[148]
Antibody-conjugated protac	ERa	VHL	Oncology: breast cancer, precision medicine	[153]
Antibody-conjugated protac	BRD4	VHL	Oncology, precision medicine	[154]
Nanobody-conjugated protac	Platform for GFP-tagged proteins or other nanobody-targetable proteins.	TRIM21 E3	Research use	[156]
Nanobody-conjugated protac	Platform for GFP-tagged proteins or other nanobody-targetable proteins.	RNF4	Research use	[159]
Protac for tagged proteins	Platform for FKBP12F36V-tagged proteins.	CRBN	Research use	[160]
Lytac	EGFR, CD71, and PD-L1	E3-independent (M6PR)	Oncology, neurodegenerative disorders	[162]
Autac	MetAP2, FK506-binding protein (FKBP12), BRD4, fragmented mitochondria,	Autophagy E3s	Oncology, neurodegenerative disorders, diabetes	[164]
Attec	mHTT	E3-independent (LC3)	Neurodegenerative disorders	[166]
Nanobody-conjugated protac	Validated against GFP-fused substrates and proliferating cell nuclear antigen (PCNA)	Slmb, a Drosophila melanogaster E3.	Research tool	[167]
Protac	SRC-1	UBR	Oncology: metastasis	[169]
Allosteric modulator	Estrogen receptor	ND (indirectly, CHIP)	Oncology: breast cancer	[134,136]

* Therapeutic potential as reported in the corresponding reference.

## Figures and Tables

**Figure 1 molecules-25-05956-f001:**
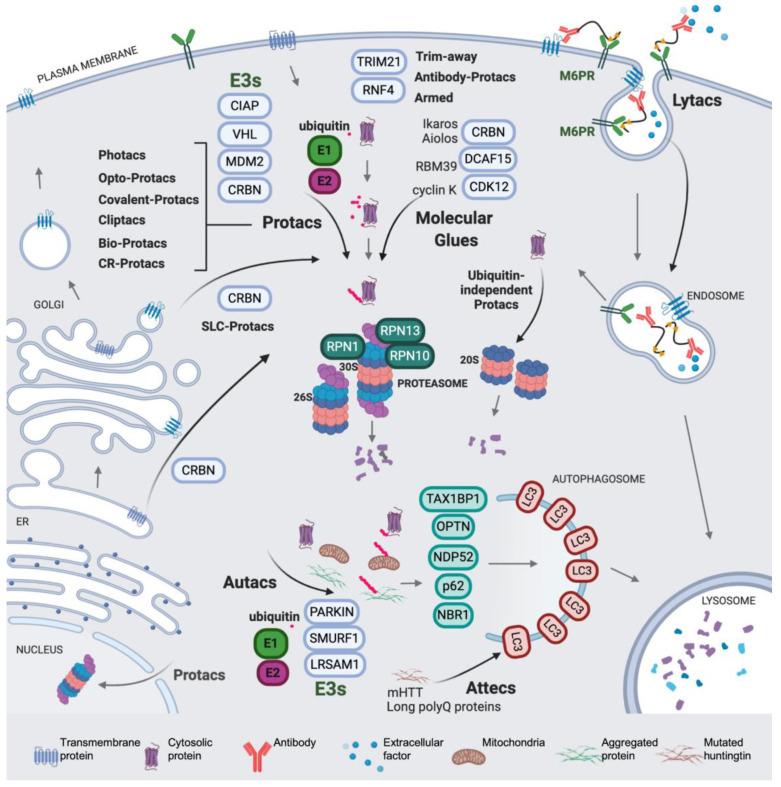
Schematic representation of the main routes of TPD in the cell. Degradation pathways targeting cargos to the proteasome and to the lysosome are shown. Curved arrows indicate the action of degrader molecules, including distinct types of protacs, molecular glues, lytacs, autacs and attecs, as they are described in the literature. Protacs and molecular glues utilize the E3 ligases shown (blue ovals), directing to the 26S and 30S proteasomes the ubiquitinated targets, which are recruited by means of the Rpn1, Rpn10 and Rpn13 proteasome receptors (dark green ovals). Ubiquitin-independent protacs direct targets to the 20S protesome. Lytacs utilize the M6PR (green) receptor to internalize external proteins to pre-lysosomal compartments, and finally deliver them to the lysosome. Autacs require E3 ligases, such as Parkin, Smurf1 and Lrsam1 (blue ovals), to target ubiquitinated cargo to autophagy receptors (OPTN, NDP53, p62, NBR1 and TAX1BP1) (green ovals), which interact with LC3 to induce cargo engulfment and autophagosome formation. On the other hand, attecs directly link protein targets with LC3, promoting engulfment and autophagosome formation. Autophagosomes are eventually integrated to the lysosome, which will hydrolyze the incoming materials. For abbreviations, see main text.

**Figure 2 molecules-25-05956-f002:**
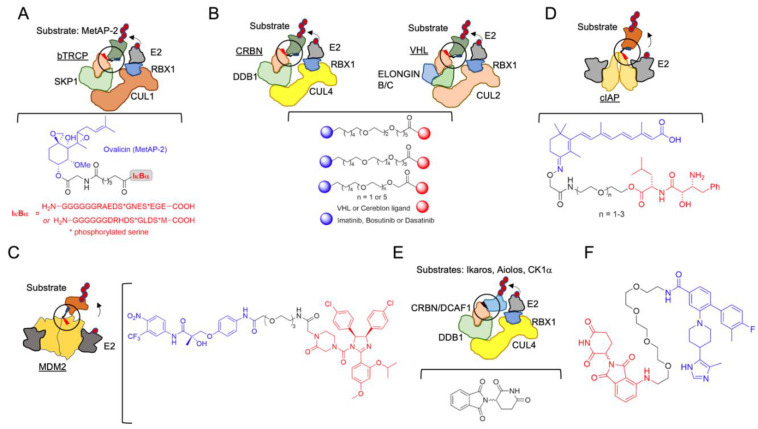
Protacs and E3 ligases. Schematic representation of the E3 ligases involved in TPD and the protac designed, as they appear in the text. RING ligases contain monomeric (e.g., PARKIN, UBR1), dimeric (e.g., cIAP, MDM2) [15,19] and multimeric forms. The Cullin RING multimeric ligase (CRL) superfamily [20] contains up to seven families, including Skp1-Cullin(Cul1)-Fbox (SCF), the DDB1-Cul4 and the elongin B, C-Cul2/5-SOCSbox protein (ECS). Thus, CRL1^TIR1^, CRL2^VHL^, CRL4^DDB1^ and CRL4^CRBN^, mentioned in the text, belong to these families [21,22,23,24]. (**A**) Structure of Protac-1, the first TPD developed. (**B**) Small-molecule protacs to target the breakpoint cluster region—Abelson tyrosine kinase (BCR-ABL). (**C**) Protac for a non-steroidal androgen receptor ligand (SARM) and the MDM2 ligand nutlin. (**D**) An example of SNIPERs to recruit the homodimeric E3 cellular cIAP1 for the degradation of retinoic acid-binding proteins (CRABP-I and II). (**E**) The structure of thalidomide and its role as degrader. (**F**) Protac d9A-2 for the degradation of SLC9A1 from leukemic cells. E3 ligase ligands are shown in red and ligands for the POIs are shown in blue. Black circles show the regions of interaction induced by the degrader molecule. Curved arrows represent the ubiquitination process by the transfer of ubiquitin moieties (red ovals).

**Figure 3 molecules-25-05956-f003:**
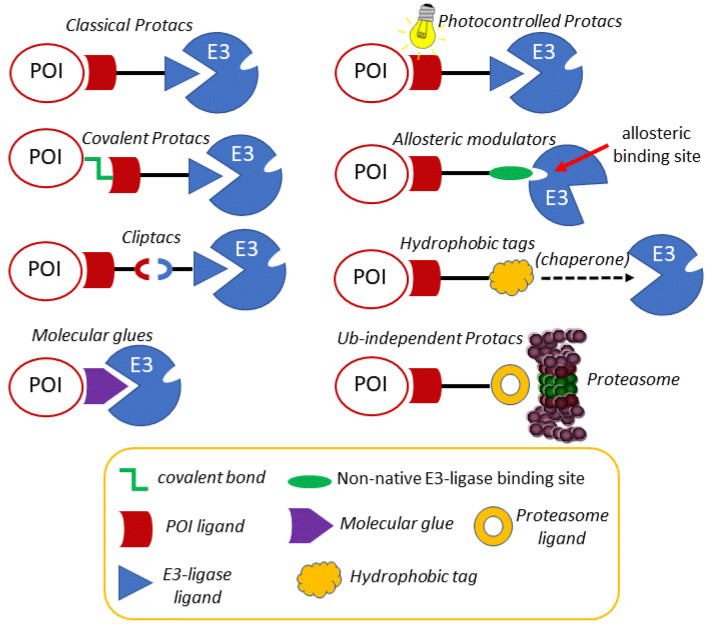
Conceptual representation of the different proteolytic chimeras described in Section 3.

**Figure 4 molecules-25-05956-f004:**
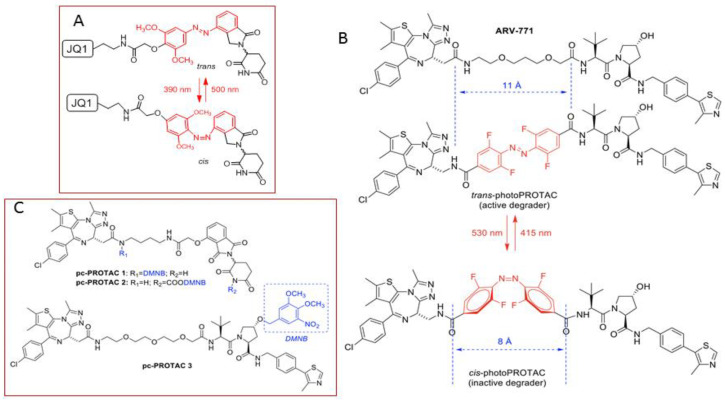
(**A**) Photac targeting the BET family of epigenetic readers BRD2-4. The photoswitchable azobenzene unit is marked in red. (**B**) Canonical protac ARV-771 and its photocontrolled version, showing the *trans-cis* isomerization between active and inactive states. (**C**) Photocaged variants of protacs developed as Brd4 degraders. In all cases, a DMNB group (in blue) is used as a phototrigger.

**Figure 5 molecules-25-05956-f005:**
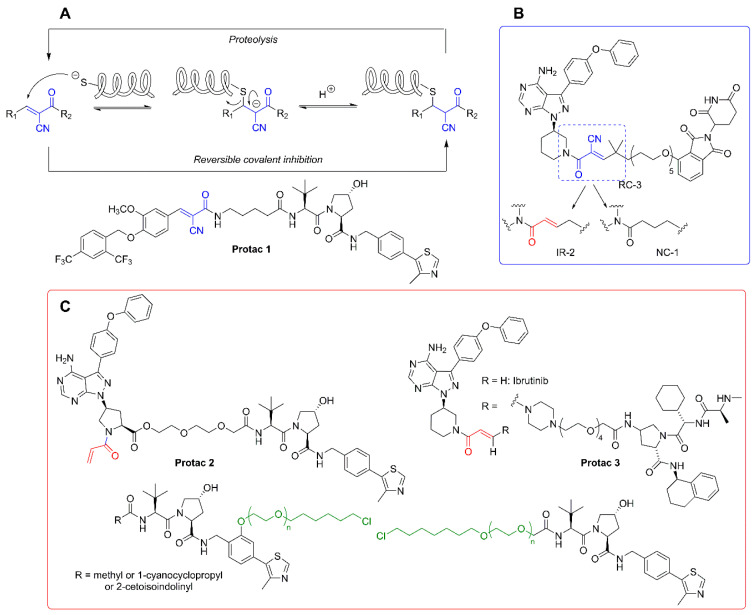
(**A**) Top: illustration of the concept of reversible covalent inhibition and application to the design of a protac against the ERRα (protac 1). (**B**) Examples of non-covalent (NC-1), irreversible covalent (IR-2) and reversible covalent (RC-3) protacs against BTK. (**C**) Examples of covalent protacs designed as BTK degraders. The Michael acceptor moiety, used as a warhead, is marked in red. Examples of HaloProtacs by functionalization of a VHL ligand with a series of ω-chlorohexyl-PEG linkers (in green).

**Figure 6 molecules-25-05956-f006:**
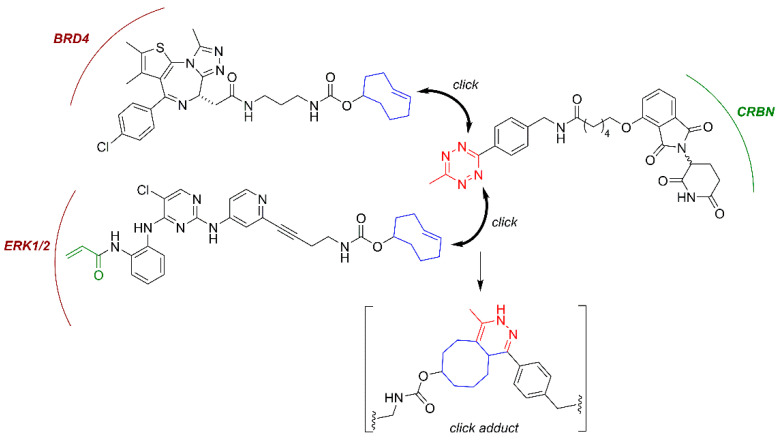
Design of cliptacs by the in-cell bioorthogonal click reaction of two smaller clickable partners. The moieties suitable for the click reaction are marked in blue and red, while the covalent warhead present in the ERK1/2 ligand is shown in green.

**Figure 7 molecules-25-05956-f007:**
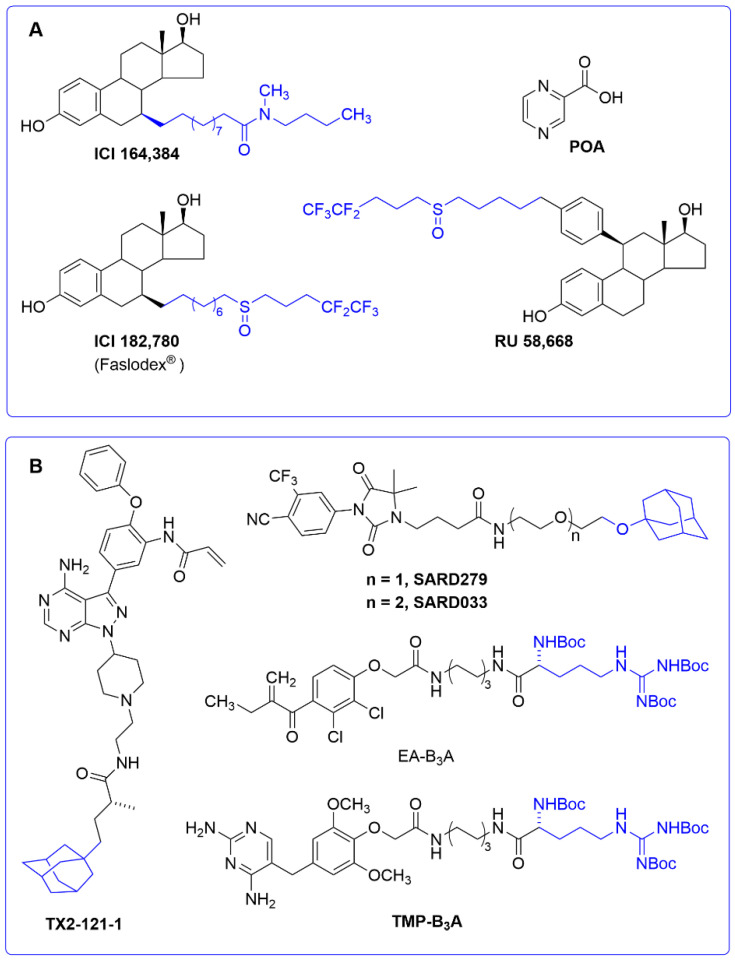
(**A**) Structures of pyrazinoic acid (POA) and selective estrogen receptor down-regulators (SERDs). (**B**) Examples of adamantyl and Boc3Arg as hydrophobic tags. Hydrophobic tags are shown in blue.

**Figure 8 molecules-25-05956-f008:**
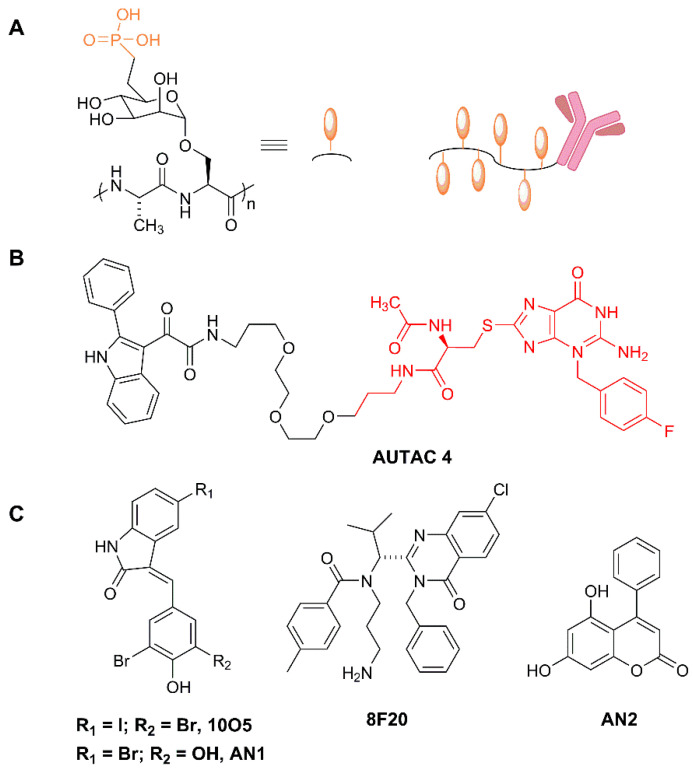
Variants of TPD directed to lysosomal degradation. (**A**) Lysosome-targeting chimeras (lytacs). Serine-O-mannose-6-phosphonate, M6Pn (left), represented as a building block to form the poly-(M6Pn) ligand attached to the POI ligand. In this example, a lytac antibody is shown (right). (**B**) Autophagy targeting chimeras (autacs). Autac 4, with the configuration 2-Phenylindole-3-glyoxyamide-PEG-p-fluorobenzyl guanine, designed to trigger mitophagy. (**C**) Autophagosome tethering compounds (Attecs). 10O5, 8F20, AN1 and AN2 act as LC3-interacting warheads for the autophagosome tethering strategy.

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
