# Peer review of "The Potential of Proteolytic Chimeras as Pharmacological Tools and Therapeutic Agents"

_molecules, 2020, doi:10.3390/molecules25245956_

Round 1

Reviewer 1 Report

My impression is that this manuscript is a outstanding review of the literature on targeted protein degradation using E3 ubiquitin ligases.  It deals with not only the potential clinical applications of PROTACs but also describes the current limitations that are only partially overcome by current technology.  The manuscript  is timely since the field of study of PROTACs is moving rapidly and can use a well structured review which helps to provide a framework for moving forward in this area. 

While I am not a biochemist I found the section on photoswitchable PROTACs exciting and fun to read.  Furthermore the review provides an up to date summary of the technology of different types of PROTACs.  

There are some minor problems with English.  Some examples below, with suggestions for correction.

line 20:  the phrase 'fate's reprogramming' is not clear.  

line 34:  the phrase 'being the proteasomes and the lysosome the two main...' should be revised  to  'proteasomes and the lysome being the two main...'

line 81:  'right productive context'  Do the authors mean 'a functional spatial context'?  

line125:  'we are in front of a new generation' .  I suggest changing to something like the following:    'it (TPD) is on the front edge of a new generation'

line 192 error in tense.  change 'were' to 'was'

line 212:  'is not acting as an inhibitor'  suggest 'does not act as an inhibitor'

line 258:  error in tense:  This sentence can be corrected by deleting 'this type of' and begining the sentence with 'Photocontrolled Protacs...'

line 275:  a similar error in tense.  again it can be corrected by deleting 'this type of' or revising the sentence.  

line 299:  again a similar error in tense.  Again this can be corrected by deleting 'This type of'

line 690: 'dwelling the'  should be 'dwelling it the'

paragraph being line 768:  Should this paragraph be subdivided into subsections 12.1 (BioProtacs), 12.2 (Conformation restricted Protacxs), 12.3 (N-degron pathway based Protacs).  , 

Author Response

We really thank the reviewer for all the comments raised and errors detected.

All the points have been fixed directly in the maintext.

Answers to points:

line 20:  the phrase 'fate's reprogramming' is not clear.  

-fixed with a more specific sentence: “With this programmed protein’s degradation , ...“ 

line 34:  the phrase 'being the proteasomes and the lysosome the two main...' should be revised  to  'proteasomes and the lysosome being the two main...'

-fixed.

line 81:  'right productive context'  Do the authors mean 'a functional spatial context'?  

-Yes, we have fixed it.

line125:  'we are in front of a new generation' .  I suggest changing to something like the following:    'it (TPD) is on the front edge of a new generation'

-fixed.

line 192 error in tense.  change 'were' to 'was'

-fixed.

line 212:  'is not acting as an inhibitor'  suggest 'does not act as an inhibitor'

-fixed.

line 258:  error in tense:  This sentence can be corrected by deleting 'this type of' and begining the sentence with 'Photocontrolled Protacs...'

-fixed.

line 275:  a similar error in tense.  again it can be corrected by deleting 'this type of' or revising the sentence.  

-fixed.

line 299:  again a similar error in tense.  Again this can be corrected by deleting 'This type of'

-fixed.

line 690: 'dwelling the'  should be 'dwelling it the'

-fixed.

Reviewer 2 Report

Coll et al. provide a comprehensive review on protactinium-like drugs. The review covers the field from the beginning to the most recent Develpments including the autophagy pathways. The review is very timely. However, the authors should consider the following points to make it more accessible for a broad readership.

  1. The illustrations focus very much on the chemical structures of the drugs. Much of the text, however, is about the conceptual differences between the different types of protactinium like drugs. At present this is not adequately covered by the figures. It would be very helpful for the reader if the authors provide molecular schemes covering the protein levels illustrating the conceptual differences for the different protactinium-like drugs throughout the manuscript.
  2. The manuscript paraphrases individual papers at length, referring a lot to the conclusions of the authors of individual papers, e.g. the section on AUTACs paraphrases almost a full page on a single paper. While it is justified to give room to major papers, the reader expects more abstraction, synthesis and reflection. This should be adapted throughout.
  3. Towards the end the writing style became more sloppy, e.g. on p.17 are many 1-sentence paragraphs and section 12 has several incomplete sentences.

Author Response

We thank the reviewer the points raised. All suggestions have been been addressed and answered.

1.The illustrations focus very much on the chemical structures of the drugs. Much of the text, however, is about the conceptual differences between the different types of protactinium like drugs. At present this is not adequately covered by the figures. It would be very helpful for the reader if the authors provide molecular schemes covering the protein levels illustrating the conceptual differences for the different protactinium-like drugs throughout the manuscript.

-A new Figure 3  emphasizing the conceptual differences among the chimeras described in the paper has been added at the beginning of Section 3 

2. The manuscript paraphrases individual papers at length, referring a lot to the conclusions of the authors of individual papers, e.g. the section on AUTACs paraphrases almost a full page on a single paper. While it is justified to give room to major papers, the reader expects more abstraction, synthesis and reflection. This should be adapted throughout.

-All paragraphs of the manuscript were written after careful reading of the cited articles. Sentences were generated in order to provide the best comprehension of the concepts and results of the works, to readers. If there's coincidence in how some aspects are exposed in the review with respect to the commented papers it is for correctness sake. Nevertheless, to avoid giving the impression of “paraphrasing”, we have corrected and improved some sentences, but we did not rewrite the whole paragraph. We thank the reviewer to point this out.

3.Towards the end the writing style became more sloppy, e.g. on p.17 are many 1-sentence paragraphs and section 12 has several incomplete sentences.

-The writing in the last sections has been thoughtfully revised in order to maintain coherence with the rest of the text. However, extending and linking some concepts in the review could hinder comprehension of the text.

Reviewer 3 Report

Ref: MOLECULES-997597
Title: The potential of proteolytic chimeras as pharmacological tools and therapeutic agents
Journal: Molecules

The article entitled: “The potential of proteolytic chimeras as pharmacological tools and therapeutic agents” by Coll et al. is a well written review article that describes the induction of protein degradation in a highly selective and efficient way by means of druggable molecules known as targeted protein degradation (TPD) and further review the breakthroughs and recent novel concepts in this highly active discipline. The manuscript as it is written now, needs major revisions and here are my suggestions/comments that will further improve the manuscript prior acceptance for publication in this journal.

Specific major comments

-This review article related to the proteolytic chimeras mechanisms and their applications as pharmacological tools is too long. I believe the article could be shortened and perhaps restructured some points by combining some sections to be easier to the readers to follow (see suggestions below).

-Figures could be combined in a few panels (or tables); it is difficult for the readers to follow so many figures.

-More emphasis should be given in some parts to the applications in the different diseases and the related pharmacological targets with some extra added short paragraphs, since this is one of the main topics of this review (please see below suggestions).

Specific minor comments:

1)Abstract:

Line 22: Please replace the “in many labs” with the “in many studies”

 2)Introduction

-lines 118-120: “In the last decades, the possibility of exploiting the cellular proteolytic systems for the design of innovative drugs has attracted the attention of researchers, since it offers the opportunity to overcome some of the limitations of classical pharmacology”: please explain further this statement by giving some examples.

-lines 137-138: “This feature opens the door to effective doses at the nanomolar range, with the positive consequences in off-target and toxicity alleviation, and in multi-drug tolerance”: please expand a bit more and explain especially the multi-drug tolerance consequence using relevant references.

-Introduction should be short; so the general overview could be used as an introduction and the other sections could be indicated separately.

3) First steps in TPD:

General comment: the title could change to “The beginning of TPD”. Also, this part could be more brief by summarizing the main findings.

-line 187: Remove the “was published” since it is redundant.

-line 189: “bestatin for POI…”: please define the word POI

-line 193: “ER lpha”: Please write the name correctly as “ER alpha”

lines 202-211: “Thalidomide was proved teratogenic…… however also immunomodulatory drug later on”: could you please discuss the effect of concentration in the immunomodulatory vs lethal mode of action for thalidomide?

4) Protacs for solute carrier proteins (SLC-Protacs)

-line 239: “Please correct the “e.gr.”

-Figure legend 2: Correct the line 243: “if the E3 ligases" to “of the E3 ligases”.

5) Photocontrolled Protacs

-lines 264-272: “The physical phenomenon that underlies the use of azobenzene as photoswitch is the possibility of light promoted trans-cis isomerization”….and other biomolecules”: you have explained very nicely the mechanism, however references should have been added in this paragraph to support the notion.

-lines 281-283: “These results were corroborated by a parallel light dependence degradation of the target BET proteins BRD2-4, as revealed by Western blot analysis”: Please add the reference.

-line 310: “the said biomolecule” please find another word to substitute the word “said”

6) Covalent Protacs

-lines 391-394: “Thus, by producing a construct  consisting of an anti-GFP nanobody conjugated to the HaloTag, a robust degradation of a GFP-tagged POIs is observed only upon treatment of the cells with the corresponding VHL-HaloProtac”: please explain briefly the biological significance and mechanism. Which type of cells have been used? Also elaborate about the use of nanobody in terms of specificity.

-Figure-9: the content is very limited for being in a separate figure. It would be good if you could please consider combining the chemical structure-based figures in one figure with different panels (i.e A, B, C panel etc).  This would help the readers and would significantly reduced the amount of figures.

7) In cell click-based Protacs (CLIPTACs)

-Figure 10: again could be in one panel with the other structures, if possible.

For instance the different PROTACS types could be combined in one figure.

8) Ubiquitin-independent Protacs

-General comment: A reference to a figure within the text would help explain this mechanism. It could be incorporated in one of the already given figures or as a separate figure.

9) Antibody-Chimeric Degrader Conjugates

-lines 541-543:The authors then explored the possibility of using antibodies and drug conjugation technology originally intended to deliver cytotoxic payloads to the cell to deliver Protacs”: please add a reference here to support the statement.

-line 547: “it has been described as a tested as a means….”: please rephrase

-lines 550-553: you need to add the reference(s) to support this paragraph.

-lines 575-589: Maniero et al study should have been described briefly, just summarizing the main findings.

10) Lysosome Targeting Chimeras (Lytacs) for endocytically internalized Targets

-General comment: Again this part could be shorter by only emphasizing the main findings.

-lines 638-640: “In a brilliant design, Banik et al. take advantage of this endogenous mechanism to develop a degradative tool with promising high-efficiency applications in cancer, neurodegeneration and multiple additional diseases.”:  more emphasis should be given here to the applications of the different diseases

11) Autophagy Targeting Chimeras (Autacs)

-Again you could summarize or simply mention the old-version concepts and explain more the recent studies.

-lines 729-730:”… probably because small-sized organelles facilitated the engulfment process and subsequent autophagosome formation.”: could you please explain more this statement?

-line 743: “Autac4 or improved derivatives, as a drug”: would be better to be written as “for drug development”

12) Autophagosome Tethering Compounds (Attecs)

-lines 738-739: “….mitochondria, followed by regeneration of the organelle. This important healing effect of Autac4 has therapeutic applications in degenerative pathologies”: Please explain more the impact that this mechanism may have in neurodegenerative or other diseases.

13) Figure 14: likewise to my previous comments for other figures; it should be incorporated in one panel with others.

14) Miscelaneous Protacs

-Clear subheadings should be included in this part (i.e line 771, 778).

-line 787: “Authors test the N-degron LRAA-YL2 Protac in several cancer cell lines and observe efficient”: please replace the “test” with “tested” and the “observe” with “observed”

15) Concluding remarks: an exciting third generation of protein degrading chimeras

-Line 798: “… approach to create new drugs with improved capacity of inactivating disease-related proteins”: maybe would be better to use the term “proteolytically inactivated disease-related proteins”.

-A small paragraph emphasizing the therapeutic potential of the technologies in different diseases with examples would put additional value in this section.

16) Table 1: -It would be really useful if you could include an additional column related to the therapeutic potential in different diseases  summarizing the applications that have been discussed in the text.

-General comment: Please check the numbers/order of the references in the table (i.e Ref 69, 70 etc) and throughout the text and make sure that have been allocated correctly.

Author Response

-We thank the reviewer for the comments and suggestions. We have addressed these important points in the next sections. As shown in more detail below, we have shortened and reorganized the manuscript, grouped figures (Figures 3-5, 6-9, and 11-12 have been combined into the new Figures 4, 5, and 7, respectively; in addition, Figure 14 has been deleted) and put more emphasis on therapeutic applications, as requested.

 In general, we have rather deleted the conflictive sentences than extended the arguments, due to the high amount of information contained in the manuscript and the need of reducing its size.

Specific minor comments:

1)Abstract:

Line 22: Please replace the “in many labs” with the “in many studies”

-We have applied this correction.

 2)Introduction

lines 118-120: “In the last decades, the possibility of exploiting the cellular proteolytic systems for the design of innovative drugs has attracted the attention of researchers, since it offers the opportunity to overcome some of the limitations of classical pharmacology”: please explain further this statement by giving some examples.

-Examples of this statement were given in the next paragraph. In order to make these important concepts more comprehensible, we have put together both paragraphs and adapted sentences. Moreover, we have included the reference Burslem et al, Cell Chemical Biology 25, 67–77, January 18, 2018, in which the performances of a classic inhibitor and of a protac towards the same target  are compared.

-lines 137-138: “This feature opens the door to effective doses at the nanomolar range, with the positive consequences in off-target and toxicity alleviation, and in multi-drug tolerance”: please expand a bit more and explain especially the multi-drug tolerance consequence using relevant references.

-we have removed “and in multi-drug tolerance” because it was certainly misleading, being used the term as “multi-drug combination” rather than as “multi-drug tolerance”.  We think it is more correct to use “...positive consequences in off-target and toxicity alleviation.” because it conceptually links better with the notion of low drug concentration doses.

Introduction should be short; so the general overview could be used as an introduction and the other sections could be indicated separately.

-The introduction has been shortened. Following the referee’s recommendations, the General Overview is now the Introduction and the old subsection 1.2 (“First steps in TDP”) has turned into Section 2 (renamed as “The beginning of TDP”).

3) First steps in TPD: 

General comment: the title could change to “The beginning of TPD”. Also, this part could be more brief by summarizing the main findings.

line 187: Remove the “was published” since it is redundant.

-fixed.

line 189: “bestatin for POI…”: please define the word POI

-fixed.

line 193: “ER lpha”: Please write the name correctly as “ER alpha”

-fixed.

lines 202-211: “Thalidomide was proved teratogenic…… however also immunomodulatory drug later on”: could you please discuss the effect of concentration in the immunomodulatory vs lethal mode of action for thalidomide?

-The thalidomide tragedy produced birth defects in thousands of children during the 60s. Since then, the drug is strictly prohibited to pregnant women, existing a very strict protocol for administration to women at the present. Even more important than drug concentration are the critical exposure periods for thalidomide embryopathy during human development. Most defects are associated with a critical exposure period from 21-36 days post-fertilization. After the 60s, thalidomide, when administered to other population groups, has not caused major lethality so far. we have introduced tha sentence: "These drugs, being strictly prohibited to pregnant women in order to prevent embryopathy, can be administered with side but no lethal effects in specific clinical cases (Ref: Kim and Scialli, Toxicological Sciences 122(1), 1–6, 2011)".

4) Protacs for solute carrier proteins (SLC-Protacs)

line 239: “Please correct the “e.gr.”

-Fixed.

Figure legend 2: Correct the line 243: “if the E3 ligases" to “of the E3 ligases”.

-Fixed.

5) Photocontrolled Protacs

lines 264-272: “The physical phenomenon that underlies the use of azobenzene as photoswitch is the possibility of light promoted trans-cis isomerization”….and other biomolecules”: you have explained very nicely the mechanism, however references should have been added in this paragraph to support the notion.

-These references have been added to this section. 

Sekkat, Z. Photo-Orientation by Photoisomerization. In Photoreactive Organic Thin Films; Sekkat, Z., Knoll, W. B. T.-P. O. T. F., Eds.; Elsevier: San Diego, 2002; pp 63–104.

Pfaff, P.; Samarasinghe, K. T. G. G.; Crews, C. M.; Carreira, E. M. Reversible Spatiotemporal Control of Induced Protein Degradation by Bistable PhotoPROTACs. ACS Cent. Sci. 2019, 5 (10), 1682–1690.

lines 281-283: “These results were corroborated by a parallel light dependence degradation of the target BET proteins BRD2-4, as revealed by Western blot analysis”: Please add the reference.

-Reference added:

Reynders, M.; Matsuura, B. S.; Bérouti, M.; Simoneschi, D.; Marzio, A.; Pagano, M.; Trauner, D. PHOTACs Enable Optical Control of Protein Degradation. Sci. Adv. 2020, 6 (8), eaay5064.

line 310: “the said biomolecule” please find another word to substitute the word “said”

-Rephrased as requested: A biomolecule is regarded as photocaged if the removal of the protecting group (or uncaging) is carried out by light

6) Covalent Protacs

lines 391-394: “Thus, by producing a construct  consisting of an anti-GFP nanobody conjugated to the HaloTag, a robust degradation of a GFP-tagged POIs is observed only upon treatment of the cells with the corresponding VHL-HaloProtac”: please explain briefly the biological significance and mechanism. Which type of cells have been used? Also elaborate about the use of nanobody in terms of specificity.

-The sentence has been modified in this way: “Thus, by producing a construct consisting of an anti-GFP nanobody (aGFP) conjugated to the HaloTag, a robust degradation of a GFP-tagged POIs is observed only upon treatment of a variety of cells (A549, ARPE-19, HEK293, HEK293-FT, and U2OS) with the corresponding VHL-HaloProtac [122]. In this case, an antigen-stabilized aGFP mutant, only stable when bound to the antigen, was used to increase the specificity of the degradation machinery. This paper also illustrates the efficiency of camelid derived nanobodies as robust tools for selective target recognition, despite the requirement of rather elaborated POI-GFP and Halo-aGFP constructs [123]”

Figure-9: the content is very limited for being in a separate figure. It would be good if you could please consider combining the chemical structure-based figures in one figure with different panels (i.e A, B, C panel etc).  This would help the readers and would significantly reduced the amount of figures.

-Figures 3-5, 6-9, and 11-12 have been combined into the new Figures 4, 5, and 7, respectively. In addition, Figure 14 has been deleted.

7) In cell click-based Protacs (CLIPTACs)

Figure 10: again could be in one panel with the other structures, if possible.

For instance the different PROTACS types could be combined in one figure.

-A new Figure 3, at the beginning of Section 3 has been added to illustrate the different types of Protacs described in the Section.

8) Ubiquitin-independent Protacs

General comment: A reference to a figure within the text would help explain this mechanism. It could be incorporated in one of the already given figures or as a separate figure.

-A mention to Figure 3 has been included in the text.

9) Antibody-Chimeric Degrader Conjugates

lines 541-543:The authors then explored the possibility of using antibodies and drug conjugation technology originally intended to deliver cytotoxic payloads to the cell to deliver Protacs”: please add a reference here to support the statement.

-Reference added: Jiang 2018;

line 547: “it has been described as a tested as a means….”: please rephrase

-Rephrased as follows: CLL1 is overexpressed in AML-related cells [152] and has been validated as an antigen for the delivery of Antibody-Drug conjugates (ADCs) to acute AML cell lines [151].

lines 550-553: you need to add the reference(s) to support this paragraph.

-References added in order to support the paragraph: Coats 2019 Pike 2020.

lines 575-589: Maniero et al study should have been described briefly, just summarizing the main findings.

-This paragraph has been summarized and specific details about Maniero’s work have been omitted in order to match the level of detail used in describing other studies.

10) Lysosome Targeting Chimeras (Lytacs) for endocytically internalized Targets

General comment: Again this part could be shorter by only emphasizing the main findings.

lines 638-640: “In a brilliant design, Banik et al. take advantage of this endogenous mechanism to develop a degradative tool with promising high-efficiency applications in cancer, neurodegeneration and multiple additional diseases.”:  more emphasis should be given here to the applications of the different diseases

-We have fused this paragraph with the consecutive paragraph, and rewritten parts of them, in order to emphasize the potential application of Lytacs to different pathologies.

11) Autophagy Targeting Chimeras (Autacs)

Again you could summarize or simply mention the old-version concepts and explain more the recent studies.

-We have suppressed some parts in order to summarize. 

lines 729-730:”… probably because small-sized organelles facilitated the engulfment process and subsequent autophagosome formation.”: could you please explain more this statement?

-We have deleted this sentence because it is not necessary to speculate on the observed fact that fragmented mitochondria are degraded by Autacs. We thank the reviewer to raise this point.

line 743: “Autac4 or improved derivatives, as a drug”: would be better to be written as “for drug development”

-The entence has been rephrased according reviewer’s suggestions.

12) Autophagosome Tethering Compounds (Attecs)

lines 738-739: “….mitochondria, followed by regeneration of the organelle. This important healing effect of Autac4 has therapeutic applications in degenerative pathologies”: Please explain more the impact that this mechanism may have in neurodegenerative or other diseases.

-In the following paragraph, the specific and highly relevant case of Down Syndrome is explained and discussed. We would prefer not to add any additional example in order to avoid over-sizing the section again.

13) Figure 14: likewise to my previous comments for other figures; it should be incorporated in one panel with others.

-We have deleted the Figure, since this particular structure is not relevant to illustrate the concept. The interested reader can always check the structure in the accompanying reference. 

14) Miscelaneous Protacs

Clear subheadings should be included in this part (i.e line 771, 778).

-Fixed.

line 787: “Authors test the N-degron LRAA-YL2 Protac in several cancer cell lines and observe efficient”: please replace the “test” with “tested” and the “observe” with “observed”

-Fixed.

15) Concluding remarks: an exciting third generation of protein degrading chimeras

Line 798: “… approach to create new drugs with improved capacity of inactivating disease-related proteins”: maybe would be better to use the term “proteolytically inactivated disease-related proteins”.

-Fixed-

A small paragraph emphasizing the therapeutic potential of the technologies in different diseases with examples would put additional value in this section.

-This paragraph has been added: “From a clinical perspective, a non-negligible number of therapeutically relevant proteins have been targeted using the Protac technology. Despite only two compounds are currently in clinical trials for some varieties of resistant breast and prostate cancers, this number is expected to rise in the forthcoming years”

16) Table 1: -It would be really useful if you could include an additional column related to the therapeutic potential in different diseases  summarizing the applications that have been discussed in the text.

General comment: Please check the numbers/order of the references in the table (i.e Ref 69, 70 etc) and throughout the text and make sure that have been allocated correctly.

-The Table has been modified as requested

Round 2

Reviewer 3 Report

Ref: MOLECULES-997597-R1
Title: The potential of proteolytic chimeras as pharmacological tools and therapeutic agents
Journal: Molecules

The manuscript entitled: “The potential of proteolytic chimeras as pharmacological tools and therapeutic agents” by Coll et al. is a well written review article and all the major/minor comments and suggestions have been addressed by the authors adequately. Therefore the manuscript has been greatly improved and can be considered for acceptance for publication on this journal in the current version.

As a last minor note, please check the references since there are some duplications in the reference list (i.e Ref  96, 98 same reference).